# Physical Activity, Nutritional Behaviours and Depressive Symptoms in Women with Hashimoto’s Disease

**DOI:** 10.3390/healthcare13060620

**Published:** 2025-03-13

**Authors:** Maria Gacek, Agnieszka Wojtowicz, Jolanta Kędzior

**Affiliations:** 1Department of Sports Medicine and Human Nutrition, University of Physical Culture in Kraków, 31-571 Krakow, Poland; 2Department of Psychology, University of Physical Culture in Kraków, 31-571 Krakow, Poland; agnieszka.wojtowicz@awf.krakow.pl; 3College of Physical Education and Sport, University of Bielsko-Biała, 43-309 Bielsko-Biała, Poland; jkedzior@ubb.edu.pl

**Keywords:** hypothyroidism, lifestyle, diet, depression

## Abstract

An important element of supporting pharmacotherapy in hypothyroidism is a pro-health lifestyle, with rational nutrition and recreational physical activity playing important roles. **Objectives**: The aim of this study was to analyse selected behavioural determinants of depressive states in women with Hashimoto’s disease. **Methods**: This study was conducted among 219 women aged 20–50 using the following: (i) the author’s questionnaire of nutritional behaviours for people with hypothyroidism (QNB); (ii) the International Physical Activity Questionnaire (IPAQ); and (iii) the Beck Depression Scale (SDB). Statistical analysis was performed in Statistica 13.1 and JASP programmes, using Spearman’s R correlation analysis, the Kruskal–Wallis analysis of variance and regression analysis at a significance level of α = 0.05. **Results:** It was found that depressive symptoms were weakly negatively associated with moderate physical activity. At the same time, women with low levels of depressive symptoms demonstrated higher levels of vigorous physical activity than women with moderate and high levels of depression. Depression symptoms decreased with the implementation of some dietary recommendations, including the consumption of products rich in iodine, iron, zinc, selenium, vitamin D, vitamin A and polyunsaturated omega-3 fatty acids. The occurrence of constipation, requiring a high-fibre diet, was positively associated with symptoms of depression. Regression analysis showed that the model consisting of all QNB items explained 18% of depression symptoms. **Conclusions**: In summary, among women with Hashimoto’s disease, moderate physical activity and some rational dietary choices were associated with a lower intensity of depressive symptoms. Promoting a healthy lifestyle may help improve the mental state of patients with Hashimoto’s disease.

## 1. Introduction

Chronic autoimmune Hashimoto’s thyroiditis, the most common cause of hypothyroidism, often coexists with other autoimmune and cardiometabolic diseases and neuropsychiatric disorders, including those related to depression [1,2,3,4,5]. The prevalence of depressive disorders among people with hypothyroidism, especially in women, has been confirmed by systematic literature reviews and meta-analyses based on studies from different regions of the world [2,6,7]. Disorders of the hypothalamic–pituitary–thyroid axis play a key role in the etiopathogenesis and neurobiology of depressive disorders in the course of Hashimoto’s thyroiditis [8]. Autoimmune thyroid diseases can lead to a significant deterioration in an individual’s quality of life dependent on their health condition in terms of physical and mental functioning, especially in the state of hypothyroidism but also in the state of euthyroidism [1]. Depressive disorders are a function of interactions between genetic, psychological, biological and environmental factors [9] and, at the same time, they show diversity, and the current classification of the American Psychiatric Association DSM-5 includes eight types of them, including depressive disorders due to another medical condition [10,11]. Depressive disorders are characterized by a low mood, anhedonia, increased fatigue and disturbances in appetite and sleep, as well as frequent thoughts and actions of suicide [10,11]. They are also often associated with a sense of persistent, chronic, undefined anxiety, which further impairs daily functioning [10,11].

The etiopathogenesis of Hashimoto’s disease includes genetic predispositions and environmental factors, including those related to lifestyle [4,12,13]. In this context, a significant aspect regarding the prevention and support of pharmacotherapy in hypothyroidism is a healthy lifestyle, with a special role played by a rational diet and recreational physical activity [3,4,14,15,16,17]. Health training supports weight loss and the normalization of glycaemia and the blood lipid profile, and it also reduces psychological stress while improving mental states [18,19,20]. In research on the subject, the importance of undertaking regular physical activity has been confirmed for improving physical fitness and body composition, including increasing muscle mass in women with Hashimoto’s disease [21,22].

A varied, balanced diet, based on the canon of rational nutrition, taking the specificity of the disease into account, can also reduce inflammation and the risk of complications in hypothyroidism [4,14,15,16]. The diet should include nutrients that condition and support thyroid function, with a limit on ingredients that inhibit thyroid activity [3,4,15,23,24,25]. Potential allergies and food intolerances as well as other health problems (constipation and excess body mass) should also be considered [26,27,28]. Eating behaviours are also important within the context of levothyroxine replacement therapy effectiveness. Periodically, the so-called autoimmune protocol, referring to the restrictive paleo diet, can be used, which reduces the hyperactivity of the immune system and may lead to disease remission [29,30]. In general, an individually selected diet, especially the Mediterranean one, which is anti-inflammatory, can support thyroid function and reduce the risk of developing metabolic and mental disorders [4,15,16,28,31].

Despite the great significance of environmental factors, including behavioural ones, both in the aetiology and prevention and treatment of hypothyroidism, the limited scale of pro-health behaviours in the lifestyle of patients has been confirmed in numerous studies. Poor dietary choices and a low level of physical activity, limiting the effectiveness of therapeutic procedures, have also been demonstrated in various studies [4,32,33,34,35,36]. Attention is also drawn to the low level of knowledge on nutritional recommendations for individuals with Hashimoto’s thyroiditis and the need for effective nutritional education [37]. The literature also contains studies on the increased risk of depression in the course of Hashimoto’s thyroiditis [2].

However, an unexploited area of research encompasses the issue of behavioural determinants regarding the mental state of people with autoimmune thyroid diseases. Moreover, in order to increase the effectiveness of hypothyroidism treatment and its accompanying complications, a comprehensive diagnosis of the patient, taking mental state into account, is of key importance. In reference to the holistic concept of health and the salutogenic orientation of medical sciences, research was therefore conducted on the behavioural and psychological aspects of health among women with Hashimoto’s disease. The novelty of this study is its interdisciplinary and holistic dimension, taking behavioural and psychological aspects into account, including psychopathological ones.

The objective of the study was to analyse selected behavioural determinants of depressive states in women with Hashimoto’s disease. The following research questions were formulated: (i) How are physical activity, nutritional behaviours and depressive symptoms shaped in women with Hashimoto’s disease? (ii) What are the relationships between the level of physical activity, rational nutritional behaviours and the severity of depressive symptoms in women with Hashimoto’s disease? The research hypothesis, assuming that the severity of depression symptoms in women with hypothyroidism due to Hashimoto’s disease increases with a lower level of physical activity and less rational nutritional behaviours, was subjected to empirical verification.

## 2. Materials and Methods

### 2.1. Participants

This study was conducted among women diagnosed with hypothyroidism associated with Hashimoto’s disease and who were under the care of an endocrinologist from the southern region of Poland (Małopolska and Silesian Voivodeships). The research was conducted by the authors in direct contact with patients of endocrinology clinics. The study included a group of 219 women aged 20–50 (M = 33.8; SD = 9.9). A statistical assessment of the group size in the G*Power 3.1 programme showed that the required sample size for selected statistical analyses (correlation and regression analyses) ranged from 107 for a strong effect size (f^2^ = 0.35) to 226 individuals for a moderate effect size (f^2^ = 0.15). Thus, the study group met the criterion of the statistically expected size. The inclusion criteria for the study group included the female gender, being an adult, hypothyroidism diagnosis—associated with Hashimoto’s disease—being under the care of an endocrinologist and providing written informed consent to participate in the study. Exclusion criteria concerned failure to meet the indicated inclusion criteria. The size of the group is similar or larger than those given in several of the other cited works [1,12,28,31,34]. The data were collected between 2019 and 2022, in periods when there was no lockdown in Poland due to the COVID-19 pandemic, i.e., social contacts were not limited.

The sociodemographic characteristics allow us to determine that the group was dominated by women from cities with more than 100 thousand inhabitants (40.4%), there were fewer women from cities with 20–100 thousand residents (24.8%) and villages (23.8%), and the fewest were from cities with less than 20 thousand inhabitants (11.0%). In terms of education level, the group included the most women with higher education (77.9%) and fewer with secondary education (19.3%) and basic vocational education (2.8%).

The group was dominated by women with a normative BMI (Body Mass Index) (52.1%) and those overweight (29.7%); less were obese (12.8%) or underweight (5.4%). The average BMI value was 24.2 kg/m^2^ (SD = 6.1). Women’s somatic data were obtained based on women’s declarations. All women took levothyroxine. In terms of comorbidities with Hashimoto’s, the examined women most frequently declared having coeliac disease (6.4%), type 1 diabetes (6.4%) and irritable bowel syndrome (6.4%).

### 2.2. Research Tools

#### 2.2.1. Evaluation of Nutritional Behaviours

The dietary pattern was assessed using an original questionnaire for evaluating the nutritional behaviours of individuals with hypothyroidism (QNB, see Appendix A), which included qualitative dietary recommendations in hypothyroidism described in the literature [4,14,15,16]. The questionnaire consists of 21 items concerning various aspects of the diet recommended in hypothyroidism, using a 5-point response scale (according to the Likert scale): 1—‘definitely not’, 2—‘rather not’, 3—‘difficult to say’, 4—‘rather yes’, and 5—‘definitely yes’. Thus, the results of the questionnaire were within the range of 21–105 points and interpreted in such a way that the higher the result, the more rational the dietary behaviours. The next items of the questionnaire concerned the consumption of products rich in nutrients conditioning thyroid function (iodine, iron, zinc and selenium), the consumption of products rich in nutrients supporting thyroid function (vitamins A and D and omega-3 PUFAs), limiting the consumption of products rich in ingredients inhibiting thyroid function (goitrogenic substances in cruciferous vegetables and soy), the occurrence of additional complications, including food intolerances, constipation, excessive body mass and the use of adequate dietary modifications, and nutritional behaviours related to taking thyroxine, following the autoimmune protocol and compliance with medical recommendations. The applied questionnaire allowed us to determine the sources of the included nutrients in order to limit the possible influence of women’s lower nutritional knowledge on their declared eating habits. The questionnaire was subjected to a validation procedure and psychometric assessment, which allowed us to confirm the sufficient reliability and internal consistency of the scale. The test validity was estimated using the repeated testing method (*n* = 30). The value of the Pearson’s r linear correlation coefficient was calculated and the null hypothesis test H0: r = 0 was performed using Student’s *t*-test. The result was r = 0.39 (*p* = 0.04). The reliability of the QNB questionnaire was additionally verified using a nutritional method encompassing a quantitative assessment of nutrient intake among 31 women based on the ongoing note-taking method (three-day food diaries) via the Aliant programme. A comparison of the results regarding the QNB questionnaire (declared nutritional behaviours) with the results of nutrient intake (iodine, iron, zinc, selenium, vitamin D, vitamin A, PUFAs and fibre) showed significant positive correlations between these variables, i.e., the regular consumption of products rich in specific nutrients increased the supply of these nutrients in the diet (Spearman’s R correlation coefficients ranged from 0.68 to 0.90, *p* < 0.001). This allowed for dietary confirmation concerning the reliability of the QNB questionnaire as a tool for the reliable assessment of the nutritional choices among women with Hashimoto’s disease. Cronbach’s α coefficient was 0.82, and the average correlation between the questionnaire items was r = 0.20.

#### 2.2.2. Evaluation of Physical Activity Level

The International Physical Activity Questionnaire (IPAQ) was used to measure the level of physical activity. It allowed us to assess the level of total physical activity in four categories: vigorous activity (above 1500 or 3000 MET-min/week) and moderate activity (600–1500 or 600–3000 MET-min/week) as well as walking (below 600 MET-min/week) and sitting. The IPAQ demonstrated sufficient internal consistency because the value of Cronbach’s α coefficient was >0.70 [38].

#### 2.2.3. Evaluation of Depressive States

The Beck Depression Scale (SDB) was used to assess the level of depression. It consists of 21 multiple-choice questions (possible answers are scored from 0 to 3, i.e., from ‘no symptoms’ to ‘strong symptom’). The questions refer to various symptoms of depression (low mood, pessimism, a sense of failure, the lack of satisfaction or pleasure, a sense of deserving punishment, a sense of guilt, a negative attitude towards oneself, low self-esteem, self-destructive behaviours, tearfulness and irritability, social withdrawal, indecisiveness, low energy, a low sense of one’s own attractiveness, sleep disorders, a sense of fatigue, changes in appetite, weight loss, focusing on one’s own ailments, and a loss of libido). In the interpretation of the results, it is assumed that 0–10 points indicate ‘no depression or low mood’; 11–27 suggests ‘moderate depression’; and 28 or more points indicate ‘severe depression’ [39,40].

The research protocol was approved by the Bioethics Committee of the District Medical Chamber in Kraków (No. 102/KBL/OIL/2019, dated 2 April 2019).

### 2.3. Statistical Analyses

Statistical calculations were performed in the Statistica 13.1 and JASP 0.18.2.0 programmes. The analysed variables were described using basic statistics (the mean, median, standard deviation, minimum and maximum, and lower and upper quartiles). Due to the fact that the distribution of variables deviated from that normal, the median was adopted as a measure of central tendency. In the statistical analysis (to determine correlations between variables), Spearman’s non-parametric R correlation analysis was used. The Kruskal–Wallis non-parametric analysis of variance test was also used to determine differences in the level of physical activity and pro-health eating behaviours depending on the level of depressive symptoms. Regression analysis was also performed to determine the significance of models explaining the level of depressive symptom intensity in the studied group of women. A significance level of α = 0.05 was adopted for statistical analyses.

## 3. Results

### 3.1. Hashimoto Level of Physical Activity, Rational Nutritional Behaviours and Depressive Symptoms in Women with Hashimoto’s Disease

Among the physical activity domains (IPAQ), women with Hashimoto’s thyroiditis had the highest score for walking (Me = 1155 MET-min/week), and the median total physical activity level was 2133 MET-min/week.

The median scores in the rational eating behaviours in hypothyroidism (QNB) were 66.0, and the median scores in the Beck Depression Scale (SDB) were 10.0 (Table 1).

Among the qualitative dietary recommendations for people with hypothyroidism, women most often limited soy and soy-based products in their diet, maintained a 30–60 min break between medication and meals and followed the doctor’s dietary recommendations (Me = 5.0). To a high degree (Me = 4.0), they consumed products rich in iron, zinc, selenium, vitamin D and A, and omega-3 PUFAs and maintained a break between taking medication and supplements (Me = 4.0). At the ‘difficult to say’ level (Me = 3.0), they incorporated iodine sources in their diet, limited cruciferous vegetables and maintained a break between drinking coffee/strong tea and taking medication. They relatively did not follow a lactose-free, high-fibre or reduction diet (Me = 2.0). They definitely did not avoid dairy products due to casein intolerance and did not follow a gluten-free diet, paleo diet or the autoimmune protocol (Me = 1.0) (Table 2).

The categorization of the results on the Beck Depression Scale (SDB) allowed us to show that the study group was dominated by women without symptoms of depression or with a low mood (*n* = 118; 53.88%), and there were fewer women with moderate (*n* = 84; 38.36%) and severe depression (*n* = 17; 7.76%). It was found that women with low depressive symptoms demonstrated higher levels of vigorous physical activity than those with moderate and high levels of depression (*p* = 0.034). The remaining differences did not reach the level of statistical significance (Table 3).

### 3.2. Physical Activity Level and Depressive Symptoms in Women with Hashimoto’s Disease

It was noted that with a decrease in the level of moderate physical activity, women experienced an increase in the intensity of depression symptoms (*p* = 0.033). The remaining correlations did not reach the adopted level of statistical significance (Table 4).

Regression analysis (dependent variable—SDB and predictors—IPAQ) showed that the full model consisting of all analysed types of physical activity was not statistically significant; it explained 2% regarding the variance of depressive symptoms in the study group (Table 5). A significant positive predictor turned out to be the time devoted to sitting (Table 6).

### 3.3. Nutritional Behaviours and Depressive Symptoms in Women with Hashimoto’s Disease

In the studied women, symptoms of depression decreased along with the increase in the level of implementing the following recommended nutritional behaviours: making sure to consume sources of iodine (*p* = 0.017), iron (*p* = 0.001), zinc (*p* < 0.001), selenium (*p* = 0.011), vitamin D (*p* = 0.007), vitamin A (*p* = 0.001) and omega-3 polyunsaturated acids (*p* = 0.004). At the same time, it was observed that the occurrence of constipation, requiring a high-fibre diet, was positively associated with depressive symptoms (*p* = 0.020). The remaining relationships (the general QNB index and individual items) did not reach the assumed level of statistical significance (Table 7).

Regression analysis (dependent variable—SDB and predictors—QNB) showed that the complete model consisting of all analysed dietary behaviours was statistically significant and explained 18% of the variance of depressive symptoms in the study group (Table 8). A high-fibre diet associated with constipation turned out to be a significant positive predictor, while the regular consumption of zinc-rich products was a significant negative predictor (Table 9).

## 4. Discussion

In the discussed research study, average levels of physical activity and proper nutritional behaviours were shown, as well as a low level of depressive symptoms (or a slight decrease in mood) in women with Hashimoto’s disease. At the same time, however, a high percentage of women with moderate depression was found (approx. 38%). Moreover, significant correlations were demonstrated between the analysed aspects of lifestyle and the severity of depressive symptoms, allowing us to partially positively verify the adopted research hypothesis that the severity of depression symptoms in women with Hashimoto’s disease increases with a lower level of physical activity and less rational nutritional behaviours.

### 4.1. Physical Activity in Women with Hashimoto’s Disease

In terms of the level of physical activity, it was shown that among its domains, women with Hashimoto’s disease took up walking to the highest extent, and the level of total physical activity was equal to 2133 MET-min/week, which, in light of the recommended interpretation of the results of the IPAQ [38], allows us to assess the physical activity of the examined women as average and sufficient. However, the results regarding the physical activity level should be interpreted with some caution due to the self-report nature of the IPAQ. The results obtained in this regard should be assessed positively because physical activity should be an important element of a healthy lifestyle and is important in people with autoimmune thyroid diseases. It has a positive effect on inflammatory processes in terms of reducing the activity and concentration of inflammatory mediators, and it also promotes weight loss while improving the mental state, insulin sensitivity of tissues and blood lipid profile, which are significant aspects of health in hypothyroidism [18,19,20,21,22].

In other Polish studies, limited participation in health training was also found among individuals with Hashimoto’s disease, who usually took up physical activity 3–4 times a week (34%) and less often, most frequently in the form of going to the gym (57%) and fitness classes (43%). After the diagnosis of the disease, the studied patients increased their physical activity (45%) or maintained it at an unchanged level (47%), and they reduced it less frequently [34]. In studies conducted in the Netherlands, it has been shown that women taking levothyroxine preferred high- rather than moderate-intensity activity. At the same time, female patients with Hashimoto’s disease reported greater limitations related to physical activity than women with hypothyroidism of a different aetiology [41].

### 4.2. Nutritional Behaviours in Women with Hashimoto’s Disease

In terms of dietary recommendations for people with hypothyroidism described in the literature [4,14,15,28], the examined women with Hashimoto’s disease most frequently declared maintaining an appropriate break between taking levothyroxine and breakfast (approx. 30–60 min), maintaining a break of several hours between taking the drug and supplements, limiting soy and soy-based products in their diets, and consuming dietary sources of ingredients that condition thyroid function (iron, zinc, and selenium) and support thyroid function (vitamins D and A and omega-3 PUFA). They followed the recommendations regarding the inclusion of dietary iodine sources (a factor in the synthesis of thyroid hormones) and limited the consumption of cruciferous vegetables (goitrogenic substances) to an average degree. They followed other recommendations, including those regarding dietary modifications related to food intolerances and constipation, as well as the paleo diet and the autoimmune protocol, to a lesser extent, which could be related to their health condition. The results regarding women’s eating habits should be interpreted with caution due to methodological limitations related to the selection of the group and the self-report nature of the questionnaire used to assess nutritional behaviours.

The obtained results can be discussed in the context of the consumption and functional properties of specific nutrients, important in the diet of people with hypothyroidism. Thus, the limited consumption of dietary iodine sources described in the studied women with Hashimoto’s disease corresponds to the low iodine supply in about one third of another group of women with Hashimoto’s disease, with an average supply close to the EAR norm, i.e., 0.095 mg/day [32]. The high implementation of the recommendation regarding the consumption of products that are a rich source of selenium is positive within the context of the role of selenium (a component of glutathione peroxidase) in reducing oxidative stress and the inflammation of the thyroid gland [24], and the reported low concentration of selenium in the blood of people suffering from chronic lymphocytic thyroiditis is due to its low consumption and smoking. Similarly, the consumption of vitamin D declared by the studied women with Hashimoto’s disease is justified in the context of its importance for the proper functioning of the thyroid gland and reports of low vitamin D concentration in people with Hashimoto’s disease compared to healthy people [42]. Attention should also be paid to the role of vitamin D as a strong immunomodulator, influencing both thyroid immunity and osteoimmunology [17]. In research carried out in Croatia on the relationship between nutritional behaviours and vitamin D levels in people with Hashimoto’s disease, it has been shown that low levels of vitamin D in blood were associated with the frequent consumption of coffee and sweets and higher levels with frequently consuming vegetables. It has therefore been confirmed that a varied and well-balanced diet can help prevent vitamin D deficiency and improve quality of life among patients with Hashimoto’s disease with impaired metabolic balance, especially in the later stages of the disease [42].

The results of the present research regarding the dietary choices of women with Hashimoto’s thyroiditis also correspond to the trends described in other groups of people with hypothyroidism. Thus, in another Polish group of patients with Hashimoto’s thyroiditis, mistakes in nutrition were demonstrated, including, among others, limited control of the diet energy value (approx. 43%), the low consumption of vegetables and fruits (only approx. 70% daily) and the excessive consumption of sweets and confectionery (approx. 35% several times a week). At the same time, 23% of the respondents declared that they modified their diet after the diagnosis of the disease, most often in the form of the general rationalization and elimination of gluten and lactose [34]. In yet another Polish study performed among patients with hypothyroidism from another region of Poland (West Pomeranian Voivodeship), it was shown that only 25% of the participants declared that they had introduced modifications to their dietary habits after the diagnosis of the disease. At the same time, quantitative abnormalities were demonstrated, including the low intake of PUFAs, digestible carbohydrates, dietary fibre, minerals (potassium, calcium and iron) and vitamin D [43], which partially and indirectly corresponds to the results of the currently discussed study. Dietary imbalance was also described in another group of women with Hashimoto’s disease who exhibited a low intake of vitamins (E, D, and B9) [44]. Also, in subsequent Polish research (Silesian Voivodeship), a high percentage of women who did not use any dietary modifications after being diagnosed with Hashimoto’s disease (52%) was described; only 15% followed a gluten-free diet, and 5% limited their intake of simple sugars and fats [45]. In turn, Greek studies showed that dietary irregularities associated with low fruit and vegetable consumption and excess body mass increased oxidative stress in women with Hashimoto’s disease [46]. Research in the USA determined that people with hypothyroidism were characterized by two main dietary patterns, one of which was based on the consumption of processed cereal products, sugar and meat and the other on the consumption of oils, nuts, potatoes and meat with reduced fat content. The dietary pattern based on the consumption of vegetables, fruits, whole grain cereal products and dairy products was less common and could generate nutritional deficiencies [36]. Meanwhile, in Hashimoto’s disease, an anti-inflammatory diet is recommended, balancing the supply of vitamin D, iodine and selenium, which are found in products rich in both anti-inflammatory antioxidants and omega-3 PUFAs [4,15,16].

### 4.3. Depressive Symptoms in Women with Hashimoto’s Disease

In the authors’ current research, the median intensity of depressive symptoms on the Beck Depression Scale (SDB) was 10.0, which means a low intensity of depressive states (or low mood) in women with Hashimoto’s thyroiditis. This is in accordance with the accepted interpretation of the results of the SDB questionnaire [39,40]. However, descriptive statistics allow us to determine that the study group included women with varying degrees of depressive symptoms; although women without symptoms of depression predominated (approx. 54%), approx. 38% showed moderate depression and nearly 8% severe depression. In other studies, the frequent occurrence of mood disorders and depressive and anxiety states in patients with Hashimoto’s thyroiditis from different regions of the world was confirmed [1,2,6,7,47,48,49]. Also, a meta-analysis of 19 clinical studies in a group of over 35,000 subjects demonstrated a high risk of developing depressive and/or anxiety disorders in patients with Hashimoto’s thyroiditis and subclinical hypothyroidism, over 2–3 times higher than in the control group [2]. Similarly, in other trials, a higher level of depressive symptoms has been noted among euthyroid Hashimoto’s patients compared to the control group (in SDB: 7.5 vs. 5.0; *p* < 0.01) but, at the same time, this level was lower than that obtained in the present study (Me = 10). Similarly, women with Hashimoto’s disease obtained lower scores in all domains of quality of life, including mental health [1].

### 4.4. Physical Activity and Depressive Symptoms in Women with Hashimoto’s Disease

In the studied group of women with Hashimoto’s disease, significant correlations were found between physical activity and the intensity of depression, with an indication of an increase in depressive symptoms along with a decrease in the level of moderate physical activity. Both average and high severity levels of depression symptoms were associated with lower levels of undertaking vigorous physical activity among women. At the same time, however, multiple regression analysis did not confirm that physical activity significantly explained the intensity of depressive symptoms in women with Hashimoto’s disease, while the relationship between physical activity and depressive symptoms was significant but weak and may have been influenced by other unaccounted factors. The trends obtained in this study correspond, to some extent, with the results of other trials, according to which recreational physical activity with the characteristics of health training aids the reduction in depression in women. Health training affects the synthesis of endorphins, serotonin and dopamine, thus exerting analgesic and euphoric effects. Physical activity is one of the methods applied to effectively cope with psychological stress. It reduces the intensity of anxiety and depressive symptoms and promotes better sleep quality and higher self-esteem [19,20]. In studies regarding the effect of resistance training on selected aspects of health, its importance has been noted not only for improving physical fitness but also for reducing depressive symptoms among middle-aged individuals [50].

### 4.5. Nutritional Behaviours and Depressive Symptoms in Women with Hashimoto’s Disease

In the discussed research study, significant correlations have also been found between the quality of nutritional behaviours and the intensity of depression in women with Hashimoto’s disease, with an indication of a decrease in depressive symptoms with an increase in the scale of some rational nutritional choices, especially those related to including products rich in ingredients conditioning thyroid function, participating in the synthesis and metabolism of thyroid hormones (iodine, iron, zinc, and selenium), and products rich in ingredients supporting thyroid function (vitamins A and D and omega-3 PUFAs). This indicates positive relationships between key rational nutritional choices and health status, including the mental state of women with Hashimoto’s disease; however, it should be emphasized that the obtained statistically significant correlations were weak. The correlation between diet and depressive states was confirmed by the performed regression analysis, which allowed us to determine that the model consisting of all analysed nutritional behaviours explained 18% of the variance concerning women’s depressive symptoms. In this respect, a significant positive predictor was a high-fibre diet associated with impaired intestinal peristalsis, which means that the occurrence of constipation requiring a diet with an increased supply of fibre was associated with more severe symptoms of depression, which could result from troublesome intestinal motility disorders. In turn, a significant negative predictor was the regular consumption of products rich in zinc, one of the components conditioning thyroid function by participating in the structure of receptor proteins for triiodothyronine (T3). Zinc, found in white meat, eggs, pumpkin and sunflower seeds, legumes, garlic and onions, is also a co-enzyme of one of the antioxidant enzymes (superoxide dismutase). Therefore, it enhances the antioxidant and anti-inflammatory potential of the diet, important for people with chronic thyroiditis [4]. The obtained results suggest the positive significance of a diet including products rich in zinc for the holistically defined health of women with Hashimoto’s disease; however, this is not a determining factor in the severity of depression.

The significance of diet quality for mental health, including the prevention of depression, has also been described by other authors, who have pointed to the special role of the Mediterranean diet [14,22,51,52,53,54,55]. The Mediterranean diet is based on the preference for the consumption of vegetables, fruit, olive oil, whole grain products, legumes, fish, and nuts, with a low consumption of meat, especially red meat, and animal fats [52]. Such a nutritional model provides a high supply of antioxidants, including anti-inflammatory polyphenols (vegetables and fruit) and also substances that condition the synthesis and metabolism of thyroid hormones, including iodine, selenium, zinc, and iron (i.e., sea fish, legumes, nuts) and ingredients supporting thyroid function, including carotenoids and retinol, vitamin D and omega-3 PUFA (i.e., vegetables, fruit, sea fish). The association of vitamin D-rich products with lower levels of depressive symptoms demonstrated in the present research indirectly corresponds to the results of studies by other authors from various countries, including the UK, Nepal, the USA, France and others, in which the association of vitamin D deficiency with the development of depression was confirmed [56,57,58,59]. In these studies, the beneficial effect of D3 supplementation on reducing symptoms of depression in patients with chronic liver diseases [60] and in women with Hashimoto’s thyroiditis was demonstrated [31], which also indirectly reflects the results obtained in the current examination. There are also studies in which a lack of association is suggested between vitamin D deficiency and the risk and course of depression [61,62]. The Mediterranean diet is also rich in B vitamins, including B6 and folic acid, which, like omega-3 PUFAs, participate in the synthesis of neurotransmitters associated with mental state (serotonin, dopamine and noradrenaline) [14,28,63]. In general, the Mediterranean diet has a positive effect on the functioning of the immune system, the composition of the gut microbiota and redox homeostasis, exerting antioxidant, anti-inflammatory and immunomodulatory effects [14], which may indirectly explain its role in individuals with inflammatory autoimmune thyroid diseases and the trends obtained in the discussed results of the current study.

In further studies, a correlation has also been noted between diet and depression, indicating a reduction in subclinical symptoms of depression along with more rational dietary choices [64,65], which corresponds to the results of the present research. Conversely, an association has also been shown between mistakes in dietary choices, including the excessive consumption of fast food products, processed foods, rich in simple sugars and saturated fatty acids, and an increased risk of depression [66,67,68,69]. The excessive consumption of saturated fatty acids and simple sugars may increase oxidative stress and intestinal barrier dysfunction and increase the production of pro-inflammatory cytokines, which increases the risk of neuroinflammation, which, in turn, may negatively affect, among other aspects, the biosynthesis of neurotransmitters (dopamine and serotonin) and increase the risk of anxiety and depressive disorders [70,71]. The relationships between the consumption of fast food products and symptoms of depression/anxiety may be bidirectional because poor mental health may also negatively affect dietary choices [72].

### 4.6. Limitations

The limitations of the work are related to, among others, the self-reported nature of the research tools used. Despite the assessment of the reliability and validity of the original nutritional questionnaire (QNB), it cannot be ruled out that the reliability of the results obtained in the area of eating habits was lower than in the field of other analysed variables, i.e., the level of physical activity and depressive symptoms, the assessment of which used standardized tools, i.e., IPAQ and SDB. The implementation of this tool to assess nutritional behaviours in hypothyroidism in other studies requires its improvement and verification based on other dietary methods. The cognitive value of this work would also be increased by including a control group in the study and the classification of Hashimoto’s thyroiditis according to TSH values, as well as by assessing the predictive role of socio-demographic characteristics. The presented results were obtained among women mostly from large urban centres with higher education, covered by medical care; thus, they cannot be representative of all women with Hashimoto’s disease. The cross-sectional nature of this study is also a limitation. The indicated limitations restrict the possibility of generalizing the results and establishing cause-and-effect relationships. Continuing research in this area should include, inter alia, a large sample size and a broader spectrum of pro-health behaviours in one’s lifestyle (including coping with psychological stress, the use of psychoactive substances, sleep hygiene and rest). It should also cover an analysis of nutrient supply in the diet and assessment of the health quality of the diet as well as nutritional status within the context of depressive symptoms and quality of life of patients with hypothyroidism of various aetiologies, also taking other variables into account, including demographic, socio-economic and health characteristics.

## 5. Conclusions

Among women with Hashimoto’s disease, an average level of physical activity and rational dietary behaviours and a low level of depressive symptoms were found. The group was dominated by women without symptoms of depression or with a low mood, but there was a high percentage of women with depression, most often moderate and less often severe.

Significant correlations were observed between certain domains of physical activity and dietary behaviours and depressive symptoms, with moderate physical activity and some rational dietary choices (including the consumption of products rich in zinc) being associated with a lower intensity of depressive symptoms in women with Hashimoto’s disease. Both average and high severity levels of depression symptoms were associated with lower levels of undertaking vigorous physical activity among women.

It can be concluded that promoting a pro-health lifestyle, including recreational physical activity and proper nutrition, may help alleviate depressive symptoms and thus improve the mental state of patients with Hashimoto’s disease; however, the observed effect sizes in the presented research are small and other factors may be more influential in this area.

## Figures and Tables

**Table 1 healthcare-13-00620-t001:** Level of physical activity (IPAQ), rational nutritional behaviours (QNB) and depressive symptoms (SDB) among women with Hashimoto’s disease (*n* = 219, descriptive statistics).

Variables		M	Me	Min	Max	Q25	Q75	SD
IPAQ MET-min/week	IPAQ Vigorous	752.5	0.0	0	16,800	0.0	960.0	1798.9
	IPAQ Moderate	631.1	40.0	0	10,800	0.0	480.0	1523.1
	IPAQ Walking	2121.6	1155.0	0	13,860	396.0	2772.0	2838.4
	IPAQ Sitting	364.8	300.0	0	2400	120.0	540.0	365.1
	IPAQ Total	3505.2	2133.0	0	26,958	933.0	4158.0	4222.2
QNB	QNB Total (general level of proper nutritional behaviours)	66.2	66.0	29	92	58.0	74.0	12.2
SDB	Level of depressive symptoms	11.9	10.0	0	45	4.0	17.0	9.5

**Table 2 healthcare-13-00620-t002:** Level of rational nutritional behaviours (individual items of the QNB) among women with Hashimoto’s disease (*n* = 219, descriptive statistics).

Consecutive Items of QNB	M	Me	Min	Max	Q25	Q75
Regular consumption of foods rich in iodine	3.2	3.0	1.0	5.0	2.0	4.0
Regular consumption of foods rich in iron	3.8	4.0	1.0	5.0	3.0	4.0
Regular consumption of foods rich in zinc	3.9	4.0	1.0	5.0	4.0	5.0
Regular consumption of foods rich in selenium	3.77	4.0	1.0	5.0	3.0	4.0
Regular consumption of foods rich in vitamin D	3.6	4.0	1.0	5.0	3.0	5.0
Regular consumption of foods rich in vitamin A	4.1	4.0	1.0	5.0	4.0	5.0
Regular consumption of foods rich in omega-3 PUFAs	3.5	4.0	1.0	5.0	3.0	4.0
Limiting cruciferous vegetables in one’s diet	3.2	3.0	1.0	5.0	2.0	4.0
Limiting soy and processed foods in one’s diet	4.2	5.0	1.0	5.0	4.0	5.0
Following gluten-free diet (for gluten intolerance)	2.1	1.0	1.0	5.0	1.0	3.0
Avoiding dairy products (for casein intolerance)	2.0	1.0	1.0	5.0	1.0	3.0
Following lactose-free diet (for lactose intolerance)	2.4	2.0	1.0	5.0	1.0	4.0
Following reduction diet (for excess body mass)	2.5	2.0	1.0	5.0	1.0	4.0
Following high-fibre diet (for constipation)	2.3	2.0	1.0	5.0	1.0	3.0
Meal 30–60 min after taking medicine (levo-thyroxine)	4.3	5.0	1.0	5.0	4.0	5.0
Breakfast (after medication) without products rich in Ca	2.7	2.0	1.0	5.0	2.0	4.0
Coffee, strong tea 2 h after taking medication	3.1	3.0	1.0	5.0	2.0	4.0
Dietary supplements—a few hours after taking medication	3.8	4.0	1.0	5.0	3.0	5.0
Following paleo diet	1.6	1.0	1.0	5.0	1.0	2.0
Autoimmune protocol (periodic use)	1.8	1.0	1.0	5.0	1.0	2.0
Following doctor’s dietary recommendations	4.3	5.0	1.0	5.0	4.0	5.0

**Table 3 healthcare-13-00620-t003:** Level of physical activity (IPAQ) and healthy eating behaviours (QNB) depending on the level of depressive symptoms in women with Hashimoto’s disease (*n* = 219, Kruskal–Wallis test).

	SDB						H (2, *n* = 219)	*p*
	Level 0 (*n* = 118)		Level 1 (*n* = 84)		Level 2 (*n* = 17)			
	Me	SD	Me	SD	Me	SD		
IPAQ Vigorous	80.00	2071.93	0.00	826.85	0.00	2881.18	6.75	0.034
IPAQ Moderate	220.00	1266.50	0.00	1587.73	0.00	2600.33	4.98	0.083
IPAQ Walking	1072.50	2134.49	1386.00	3638.73	924.00	1797.60	3.04	0.219
IPAQ Sitting	300.00	223.58	360.00	509.41	300.00	213.44	1.56	0.459
IPAQ Total	2146.50	3920.65	2133.00	4626.66	1386.00	4272.63	0.55	0.761
QNB Total	65.00	11.65	68.00	13.35	62.00	10.69	0.41	0.814

Level 0—low level of depressive symptoms/no symptoms; Level 1—moderate level of depressive symptoms; Level 2—high intensity of depressive symptoms.

**Table 4 healthcare-13-00620-t004:** Correlations between level of physical activity (IPAQ) and depressive symptoms (SDB) in women with Hashimoto’s disease (*n* = 219, Spearman’s R correlations).

Variables	*n*	Spearman’s R	t (*n* − 2)	*p*
IPAQ Vigorous and SDB	219	−0.12	−1.79	0.075
IPAQ Moderate and SDB	219	−0.14	−2.14	0.033
IPAQ Walking and SDB	219	0.05	0.80	0.422
IPAQ Sitting and SDB	219	0.11	1.58	0.115
IPAQ Total and SDB	219	0.01	0.11	0.915

**Table 5 healthcare-13-00620-t005:** Regression analysis—SDB and IPAQ (complete model).

	Multiple-R	Multiple-R^2^	Adjusted-R^2^	df-Model	df-Residual	F	*p*
SDB	0.15	0.02	<0.001	4	214	1.25	0.289

**Table 6 healthcare-13-00620-t006:** Regression coefficients—SDB and IPAQ (complete model).

	SDB-Param.	SDB-Std. Err	SDB-t	SDB-*p*	SDB-Beta (ß)	SDB-St. Err. ß
Intercept	10.17	1.10	9.23	<0.001		
IPAQ Vigorous	−0.00	0.00	−0.60	0.549	−0.04	0.07
IPAQ Moderate	0.00	0.00	0.46	0.649	0.03	0.07
IPAQ Walking	0.00	0.00	0.96	0.337	0.07	0.07
IPAQ Sitting	0.00	0.00	1.97	0.050	0.13	0.07

**Table 7 healthcare-13-00620-t007:** Correlations between healthy nutritional behaviours and depressive symptoms in women with Hashimoto’s disease (*n* = 219, Spearman’s R correlation coefficients).

Correlated Variables	N	Spearman’s R	t (*n* − 2)	*p*
Regular consumption of foods rich in iodine and SDB	219	−0.16	−2.41	0.017
Regular consumption of foods rich in iron and SDB	219	−0.22	−3.29	0.001
Regular consumption of foods rich in zinc and SDB	219	−0.24	−3.71	<0.001
Regular consumption of foods rich in selenium and SDB	219	−0.17	−2.56	0.011
Regular consumption of foods rich in vitamin D and SDB	219	−0.18	−2.74	0.007
Regular consumption of foods rich in vitamin A and SDB	219	−0.23	−3.49	0.001
Regular consumption of foods rich in omega-3 PUFAs and SDB	219	−0.20	−2.94	0.004
Limiting cruciferous vegetables in one’s diet and SDB	219	0.16	2.34	0.020
Limiting soy and processed foods in one’s diet and SDB	219	−0.01	−0.21	0.832
Following gluten-free diet (for gluten intolerance)	219	0.03	0.48	0.632
Avoiding dairy products (for casein intolerance) and SDB	219	0.05	0.73	0.468
Following lactose-free diet (for lactose intolerance) and SDB	219	0.08	1.16	0.247
Following reduction diet (for excess body mass) and SDB	219	0.10	1.44	0.151
Following high-fibre diet (for constipation) and SDB	219	0.16	2.34	0.020
Meal 30–60 min after taking medication (levo-thyroxine) and SDB	219	0.12	1.73	0.085
Breakfast (after medication) without products rich in Ca and SDB	219	0.02	0.31	0.760
Coffee, strong tea 2 h after taking thyroxine and SDB	219	0.02	0.28	0.782
Supplements a few hours after taking thyroxine and SDB	219	−0.04	−0.57	0.570
Following paleo diet and SDB	219	−0.04	−0.66	0.513
Autoimmune protocol (periodic use) and SDB	219	0.05	0.68	0.495
Following doctor’s dietary recommendations and SDB	219	−0.02	−0.36	0.720
QNB Total (general level of proper nutritional behaviours) and SDB	219	−0.02	−0.35	0.725

**Table 8 healthcare-13-00620-t008:** Regression analysis—SDB and QNB.

	Multiple-R	Multiple-R^2^	Adjusted-R^2^	df-Model	df-Residual	F	*p*
SDB	0.43	0.18	0.10	21	197	2.11	0.004

**Table 9 healthcare-13-00620-t009:** Regression coefficients—SDB and QNB (selected items).

	SDB-Param.	SDB-Std. Err	SDB-t	SDB-*p*	SDB-Beta (ß)	SDB-St. Err. ß
Intercept	16.26	5.41	3.01	0.003		
Regular consumption of foods rich in iodine	0.03	0.71	0.05	0.962	0.00	0.09
Regular consumption of foods rich in iron	−0.89	1.06	−0.84	0.402	−0.09	0.11
Regular consumption of foods rich in zinc	−2.67	1.19	−2.23	0.027	−0.25	0.11
Regular consumption of foods rich in selenium	−0.06	0.95	−0.07	0.947	−0.01	0.10
Regular consumption of foods rich in vitamin D	−0.78	0.72	−1.09	0.278	−0.10	0.09
Regular consumption of foods rich in vitamin A	0.06	1.10	0.05	0.959	0.01	0.10
Regular consumption of foods rich in omega-3 PUFAs	0.05	0.82	0.06	0.952	0.01	0.10
Limiting cruciferous vegetables in one’s diet	1.23	0.63	1.95	0.053	0.16	0.08
Limiting soy and processed foods in one’s diet	0.13	0.64	0.20	0.842	0.02	0.08
Following gluten-free diet (for gluten intolerance)	−0.42	0.64	−0.66	0.510	−0.06	0.10
Avoiding dairy products (for casein intolerance)	0.06	0.83	0.07	0.943	0.01	0.12
Following lactose-free diet (for lactose intolerance)	0.45	0.73	0.62	0.537	0.07	0.12
Following reduction diet (for excess body mass)	0.48	0.57	0.85	0.397	0.07	0.09
Following high-fibre diet (for constipation)	1.34	0.57	2.36	0.019	0.20	0.08
Meal 30–60 min after taking medicine (levo-thyroxine)	1.23	0.65	1.88	0.062	0.16	0.09
Breakfast (after medication) without products rich in Ca	0.14	0.57	0.24	0.811	0.02	0.08
Coffee, strong tea 2 h after taking medication	0.02	0.51	0.04	0.969	0.00	0.08
Dietary supplements—a few hours after taking medication	−0.61	0.54	−1.12	0.263	−0.09	0.08
Following paleo diet	−1.22	0.82	−1.50	0.136	−0.15	0.10
Autoimmune protocol (periodic use)	1.05	0.72	1.47	0.142	0.14	0.10
Following doctor’s dietary recommendations	−0.06	0.86	−0.07	0.944	−0.01	0.09

## Data Availability

Data are available upon reasonable request from the authors.

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
