# Peer review of "Physical Activity, Nutritional Behaviours and Depressive Symptoms in Women with Hashimoto’s Disease"

_healthcare, 2025, doi:10.3390/healthcare13060620_

Round 1

Reviewer 1 Report

Comments and Suggestions for Authors

Dear Authors

This work effectively incorporates the research question, objectives, and hypotheses. The empirical analysis of the results is adequate; however, the following suggestions should be taken into account:

  • Adjust the word count according to the journal’s standards. Articles should include a structured abstract of approximately 250 words.
  • The data collection process is unclear, particularly regarding how the data was gathered and who participated in the collection aside from the sample.
  • The literature review on the variable "Depression" is limited and lacks depth. This should be revised.
  • When introducing abbreviations, it is advisable to include their full meaning if relevant. For example, in line 119, BMI (Body Mass Index) should be defined, even though "muscle mass" is mentioned earlier in line 55.
  • The reliability and internal consistency of the IPAQ scale are not sufficiently addressed, despite a comment on its limitations.
  • Line 184 revised spelling error and line 243 “ ...making sure to consuming consuming sources of iodine...”
  • Ensure that the tables follow the journal’s formatting standards.

Best regards,

Reviewer 2 Report

Comments and Suggestions for Authors

General Recommendations: Thank you for the study. However, I think that some important points need to be seriously corrected in order for the study to be published, especially the fluency in the introduction and the aspect of this study that differs from other studies in the last section and the original value that it will contribute to the field.

This study addressed a gap in the literature by investigating the impact of physical activity and dietary behaviors on depressive symptoms in women with Hashimoto's disease. However, similar studies in this field may limit the claim to complete originality.

This research exhibits methodological limitations, including the use of self-reported data, a cross-sectional design, and the absence of a control group. These constraints may restrict the generalizability of the findings and hinder the establishment of causal relationships.

The study's design and execution generally adhere to scientific principles. The statistical methods employed were appropriate and well-selected. Nevertheless, specific weaknesses are evident, such as the low explanatory power of regression analysis and the issue of multiple comparisons. I recommend you to support your findings with more explanatory visual graphics. 

This study contributes significantly to the literature by examining the influence of physical activity and dietary behaviors on depressive symptoms in women with Hashimoto's disease. However, it has limitations, including methodological constraints and lack of specificity. Consequently, addressing these limitations in future research is advisable to enhance the potential for publication in high-impact journals.

Comments on the Quality of English Language

The paper should be checked by native speaker.

Reviewer 3 Report

Comments and Suggestions for Authors

The article addresses the relationship between physical activity, nutritional habits, and depressive symptoms in women with Hashimoto’s disease. While the study explores a relevant topic and employs validated methodological tools, several shortcomings compromise the robustness of its conclusions. The following critique highlights specific issues in writing, methodology, and result analysis, with direct references to the text.

The abstract lacks clarity and precision in presenting key findings. For example, in line 16, it states that "the intensity of depressive symptoms increased with a decrease in moderate physical activity (p=0.033)." However, the study does not establish causality, only a weak correlation, which makes this claim potentially misleading. Later, in line 19, it mentions that "women with low levels of depressive symptoms demonstrated higher levels of vigorous physical activity," but the corresponding p-value (p=0.034) is barely significant, calling into question the strength of this finding. Overall, the abstract includes too many p-values without adequately interpreting the effect size, which could lead to misinterpretations regarding the clinical relevance of the results.

In the introduction, the review of previous studies linking Hashimoto’s disease with depression (lines 35-44) is essentially a collection of references without a critical discussion of the mentioned studies. For instance, in line 38, it states that "autoimmune thyroid diseases can lead to a significant deterioration in quality of life," but it does not explain the specific pathophysiological mechanisms linking thyroid inflammation to depression. Furthermore, the mention of DSM-5 in line 44 is superficial and does not provide any significant insight into the study.

One of the most serious methodological issues is sample selection. In line 100, it is stated that participants were recruited from southern Poland and were all under the care of an endocrinologist. However, this introduces selection bias, as women with a confirmed diagnosis and medical treatment may differ significantly in lifestyle habits and psychological status compared to those not receiving medical supervision. Additionally, in line 118, it is noted that most participants had higher education (77.9%), skewing the sample toward a specific socioeconomic group and limiting the generalizability of the findings to populations with lower education and healthcare access.

The use of the Nutritional Behavior Questionnaire (QNB) is problematic. In line 128, it is mentioned that the questionnaire includes "qualitative dietary recommendations described in the literature," but it is unclear whether the questionnaire has been validated outside this study. In line 149, it is reported that the QNB's reliability, measured using Pearson’s correlation coefficient, was r=0.39, which is low and suggests that the questionnaire may not be a reliable indicator of dietary habits. Additionally, the method for evaluating the questionnaire’s validity is weak, as it was only compared against a small group of 31 women using food diaries (line 150). This raises concerns about whether the reported dietary intake accurately reflects the habits of the broader sample.

Regarding physical activity assessment, the use of the IPAQ (line 163) also has well-documented limitations. The IPAQ has been shown to overestimate physical activity compared to objective measurements such as accelerometers. This could have influenced the estimation of participants' physical activity levels, introducing bias into the results.

One of the study’s key findings is that moderate physical activity is negatively correlated with the severity of depressive symptoms (p=0.033, line 229). However, the regression analysis (line 235) shows that the complete model for physical activity explains only 2% of the variance in depressive symptoms. This suggests that while physical activity is associated with depression, it is not a strong predictor of its severity in this population. The interpretation of this finding should be much more cautious, emphasizing in the discussion that this relationship is weak and may be influenced by other unaccounted factors.

The analysis of nutritional habits presents similar issues. In line 243, it is reported that the consumption of zinc (p<0.001), iron (p=0.001), and other nutrients is associated with lower severity of depression. However, when adjusting the model in regression analysis (line 251), only zinc consumption remains a significant predictor, and diet explains just 18% of the variability in depressive symptoms. This indicates that dietary habits play a role but are not the determining factor in depression severity among women with Hashimoto’s disease. Additionally, the finding that constipation (and the associated high-fiber diet) predicts higher levels of depression (p=0.020, line 245) suggests that the relationship between diet and mood may be more complex and not necessarily linear.

Another issue in the presentation of results is the lack of stratified analyses. It is mentioned in line 219 that 7.76% of participants had severe depression, but there is no examination of whether the observed relationships between diet, physical activity, and depression differ based on symptom severity. A more detailed analysis could have identified subgroups of patients who benefit more from specific lifestyle changes.

The discussion largely repeats the findings without a sufficiently critical analysis. For example, in line 274, it is stated that physical activity has a "positive effect on mental state," but the study’s results only show a weak correlation. In line 320, it is mentioned that "vitamin D deficiency is associated with depression," but it does not discuss the mixed evidence in scientific literature, where some studies have found no clear relationship between vitamin D supplementation and improved mood.

The study’s conclusions (lines 478-492) are overly general and do not adequately reflect the limitations of the findings. It is stated that "promoting a healthy lifestyle may help alleviate depressive symptoms," but it does not acknowledge that the observed effect sizes in the study are small and that other factors may be more influential in depression among these patients.

The article explores a clinically relevant topic, but it has serious methodological and analytical limitations that compromise the strength of its conclusions. The sample selection is biased, the measurement of dietary habits lacks sufficient validation, and the statistical models explain only a small fraction of the variability in depressive symptoms. Additionally, the discussion does not critically address the study’s limitations, and the conclusions overstate the clinical significance of the results. To improve the study, a more rigorous analysis of confounding factors, a more representative study design, and greater caution in data interpretation are needed.

Round 2

Reviewer 2 Report

Comments and Suggestions for Authors

Thanks for editing.

Reviewer 3 Report

Comments and Suggestions for Authors

.